# circPTPN12/miR-21–5 p/ΔNp63α pathway contributes to human endometrial fibrosis

**Minmin Song[1†], Guangfeng Zhao[1†], Haixiang Sun[2†], Simin Yao[1], Zhenhua Zhou[1], Peipei Jiang[1], Qianwen Wu[1], Hui Zhu[1], Huiyan Wang[1], Chenyan Dai[1], Jingmei Wang[3], Ruotian Li[4], Yun Cao[1], Haining Lv[1], Dan Liu[1], Jianwu Dai[5]\*, Yan Zhou[6]\*, Yali Hu[1]\***

[1]Department of Obstetrics and Gynecology, The Affiliated Drum Tower Hospital of Nanjing University Medical School, Nanjing, China; [2]Center for Reproductive Medicine, Department of Obstetrics and Gynecology, The Affiliated Drum Tower Hospital of Nanjing University Medical School, Nanjing, China; [3]Department of Pathology, The Affiliated Drum Tower Hospital of Nanjing University Medical School, Nanjing, China; [4]Department of Laboratory Medicine, Jiangsu Key Laboratory for Molecular Medicine, Nanjing University Medical School, Nanjing, China; [5]Institute of Genetics and Developmental Biology, Chinese Academy of Sciences, Beijing, China; [6]Department of Obstetrics, Gynecology and Reproductive Sciences, Center for Reproductive Sciences, Eli and Edythe Broad Center of Regeneration Medicine and Stem Cell Research, University of California San Francisco, San Francisco, United States

**\*For correspondence:**
jwdai@genetics.ac.cn (JD);
yan.zhou@ucsf.edu (YZ);
yalihu@nju.edu.cn (YH)

[†]These authors contributed equally to this work

**Competing interests:** The authors declare that no competing interests exist.

**Abstract** Emerging evidence demonstrates the important role of circular RNAs (circRNAs) in regulating pathological processes in various diseases including organ fibrosis. Endometrium fibrosis is the leading cause of uterine infertility, but the role of circRNAs in its pathogenesis is largely unknown. Here, we provide the evidence that upregulation of circPTPN12 in endometrial epithelial cells (EECs) of fibrotic endometrium functions as endogenous sponge of miR-21–5 p to inhibit miR-21–5 p expression and activity, which in turn results in upregulation of ΔNp63α to induce the epithelial mesenchymal transition (EMT) of EECs (EEC–EMT). In a mouse model of endometrium fibrosis, circPTPN12 appears to be a cofactor of driving EEC–EMT and administration of *miR-21–5* p could reverse this process and improve endometrial fibrosis. Our findings revealed that the dysfunction of circPTPN12/miR-21–5 p/ΔNp63α pathway contributed to the pathogenesis of endometrial fibrosis.

## Introduction

Endometrial fibrosis is clinically characterized with intrauterine adhesions (IUA) and is often secondary to severe injury of endometrium including various uterine operations mainly repeated curettage. Endometrial fibrosis is the most common reason of uterine infertility (*Yu et al., 2008*; *March, 2011a*; *March, 2011b*). The mechanism of endometrial fibrosis remains unclear, which hampers the development of effective therapeutics for the disease (*March, 2011b*). Recently, our study showed that the ectopic expression of ΔNp63α, a transcription factor, in the endometria of IUA patients triggers the epithelial mesenchymal transition (EMT) of endometrial epithelial cells (EECs) (EEC–EMT) to promote endometrial fibrosis (*Zhao et al., 2020*).

In the process of EMT, post-transcriptional regulation has critical roles (*Nieto et al., 2016*). Studies showed that let-7d is downregulated in EMT of the lung epithelial cells and downregulation of let-7d in mice causes lung fibrosis (*Pandit et al., 2010*). miR-29b suppresses EMT of lung epithelial

cells and prevents mice against lung fibrosis (*Sun et al., 2019*). In the renal fibrosis, miR-34a induces EMT in renal tubular epithelial cells and knockout miR-34a ameliorates renal fibrosis in mice (*Liu et al., 2019a*). Moreover, miRNAs may function differently in the pathogenesis of fibrosis in different tissues. For example, miR-27a prevents the fibrosis of detrusor and kidney (*Wu et al., 2018*; *Zhang et al., 2018*), but promotes the myocardial fibrosis (*Ya-Se et al., 2018*). However, the role of miRNAs in the pathogenesis of endometrial fibrosis is less studied (*Li et al., 2016*).

Recent studies demonstrate that the expression and activity of miRNAs are regulated by circular RNAs (circRNAs). circRNAs are recognized as competing endogenous RNAs (ceRNAs) to sponge miRNAs to modulate the expression and activity of miRNAs and their target genes (*Li et al., 2018*; *Zheng et al., 2016*; *Cheng et al., 2019*). To date, little is known about the biological function of circRNAs in the pathogenesis of endometrial fibrosis. Since we found that the ectopic expression of ΔNp63α in EECs induces EEC–EMT and promotes endometrial fibrosis (*Zhao et al., 2020*), and ΔNp63α is regulated by multiple miRNAs (*Candi et al., 2015*; *Lena et al., 2008*; *Rodriguez Calleja et al., 2016*), we speculated whether circRNAs is involved in the pathogenesis of endometrial fibrosis by interacting with miRNAs. Therefore, we studied the expression changes of circRNAs/miRNAs in endometrial fibrosis and their regulatory relationship with ΔNp63α to favor the understanding of endometrial fibrosis.

## Results

### miR-21–5 p is downregulated in EECs and associated with EEC–EMT

To study the spectrum of changes in the levels of ΔNp63α antagonizer miRNAs, the transcriptomic alterations in the endometria of the patients with IUA were determined. We performed high-throughput sequencing analysis using endometrium samples in late proliferative phase from three severe IUA patients and three normal controls. The histopathological results of the samples used for high-throughput sequencing showed endometrial fibrosis in IUA patients (*Figure 1—figure supplement 1A*). With the significance threshold of absolute mean fold change > 2 and p-values<0.05, of 1681 detectable miRNAs in total, 56 were upregulated and 150 were downregulated in IUA patients, compared with those in normal controls (*Figure 1A* and *Supplementary file 1*). These differentially expressed miRNAs were also presented in a heat map (*Figure 1—figure supplement 1B*). The most significantly increased and decreased miRNAs were verified by quantitative reverse transcription–PCR (qRT-PCR) (*Figure 1—figure supplement 1C*).

Among these dysregulated miRNAs, 40 upregulated and 40 downregulated miRNAs with higher expression abundance were shown in *Figure 1B*. The expressive abundance of miR-21–5 p was the highest in endometria of normal controls but was significantly downregulated in IUA patients (*Supplementary file 2*). Interestingly, we found that the 3'-UTR of ΔNp63α has miR-21–5 p binding sites by prediction in Microrna.org database. Thus, we tested the expression change of miR-21–5 p ex vivo and confirmed the decrease of miR-21–5 p expression in endometria of IUA patients (*Figure 1C*), which was further proved by RNA-scope assay (*Figure 1D*). Meanwhile, RNA-scope assay showed that miR-21–5 p was mainly expressed in EECs (*Figure 1D*). Since epithelial cells often switch to matrix-producing myofibroblasts via EMT in tissue fibrosis process (*Iwano et al., 2002*; *Zeisberg et al., 2007*) and we proved recently that ΔNp63α inducing EEC–EMT participates in endometrium fibrosis in IUA patients (*Zhao et al., 2020*), we speculated that miR-21–5 p may serve as a protector for maintaining endometrium homeostasis and preventing EECs from EMT. Therefore, we performed loss and gain of miR-21–5 p function by transfecting miR-21–5 p mimic and inhibitor into primary EECs respectively. miR-21–5 p expression increased 11 times in EECs transfected with miR-21–5 p mimic, while the expression decreased 2.5 times after miR-21–5 p inhibitor transfection (*Figure 1—figure supplement 2*). After miR-21–5 p knockdown, the expression of epithelium-specific marker, E-cadherin, was downregulated, but mesenchymal markers, N-cadherin, α-smooth muscle actin (α-SMA), and fibronectin (FN), were overexpressed in EECs (*Figure 1E*). In contrast, overexpression of miR-21–5 p markedly decreased the expression of N-cadherin, α-SMA, and FN and increased the expression of E-cadherin (*Figure 1F*).

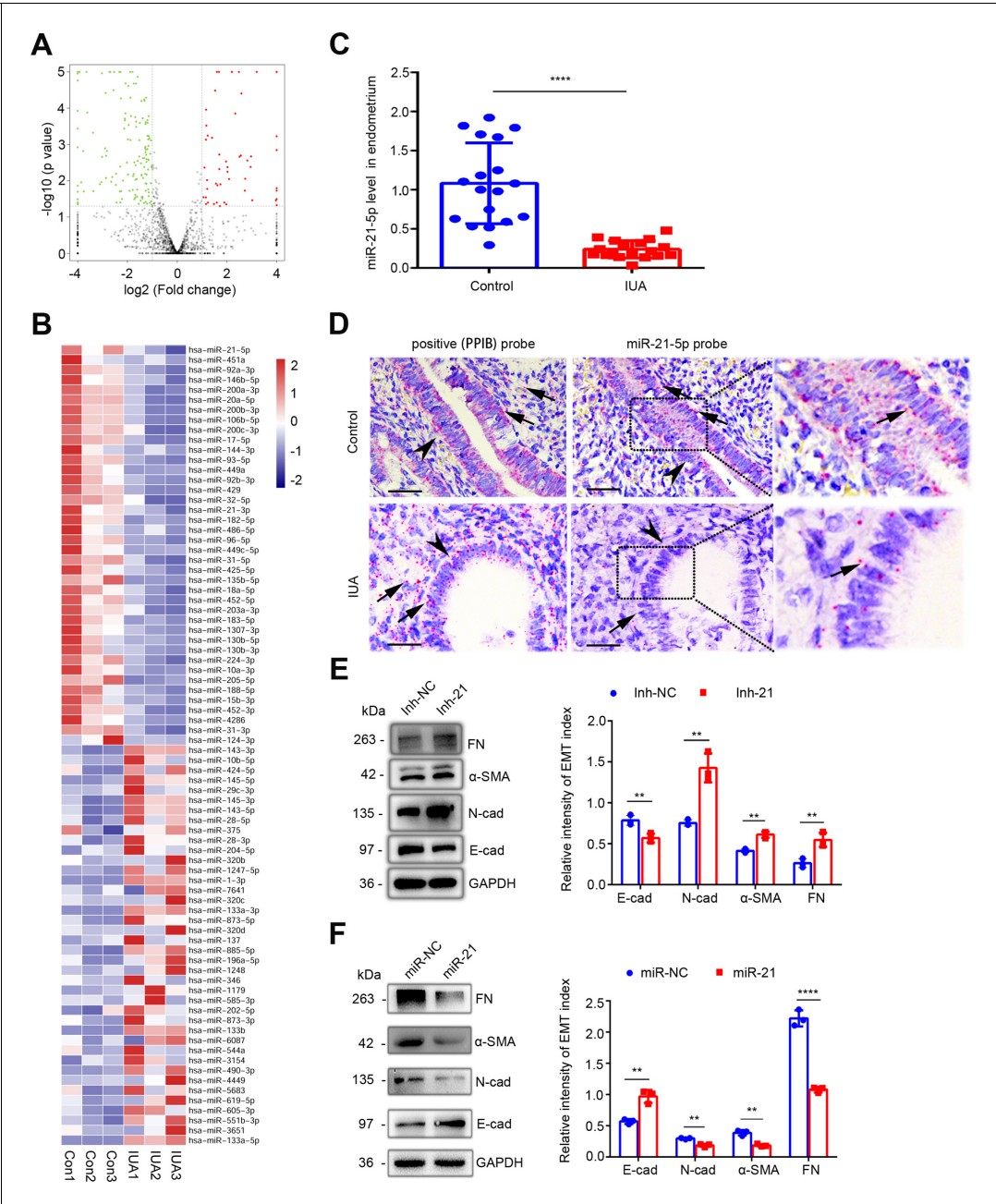

**Figure 1.** Downregulation of miR-21–5 p in endometria of IUA patients promotes EEC–EMT. (**A**) Volcano plots of miRNA-sequencing of the endometria from severe IUA patients (n = 3) and controls (n = 3) based on the high-throughput RNA-sequencing analysis. The red dots represent the miRNAs upregulated, and the green ones represent the downregulated (p<0.05 and fold change > 2). (**B**) A heatmap showing 40 upregulated and 40 downregulated miRNAs with high abundance in endometrium samples from severe IUA patients (n = 3) and controls (n = 3). (**C**) miR-21–5 p mRNA levels in endometria of severe IUA patients (n = 18) and controls (n = 18) determined by qRT-PCR. (**D**) Representative images of RNA-scope assay using specific probes to detect miR-21–5 p in the endometria of severe IUA patients (n = 3) and controls (n = 3). Peptidylprolyl isomerase B (PPIB) serves as the positive control. Arrowhead: epithelial cells; arrow (red dots): PPIB or miR-21–5 p positive dots. Scale bars 50 μm. (**E**) Left: fibronectin (FN), α-smooth muscle actin (α-SMA), E-cadherin (E-cad), N-cadherin (N-cad), and GAPDH protein levels determined by western blotting in miR-21–5 p inhibitor (inh-21)- or negative control (inh-NC)-transfected (48 hr) EECs (n = 3). Right: The quantitative band intensities determined by image J software. (**F**) Left: FN, α-SMA, E-cad, N-cad, and GAPDH protein levels determined by immunoblotting in miR-21–5 p mimic (miR-21)- or negative control (miR-NC)-transfected (48 hr) EECs (n = 3). Right: The quantitative band intensities determined by image J software. The error bars in (**C**), (**E**), and (**F**) indicate mean ± SD. **p<0.01, ****p<0.0001.

The online version of this article includes the following source data and figure supplement(s) for figure 1:

**Source data 1.** qRT-PCR data for miR-21–5 p relative expression.

*Figure 1 continued on next page*

*Figure 1 continued*

**Source data 2.** Uncropped western blots for *Figure 1E*.
**Source data 3.** The quantitative band intensities for *Figure 1E*.
**Source data 4.** Uncropped western blots for *Figure 1F*.
**Source data 5.** The quantitative band intensities for *Figure 1F*.
**Figure supplement 1.** miRNAs expression profile.
**Figure supplement 2.** miR-21–5 p transfection efficiency.

## circPTPN12 is upregulated in EECs and negatively correlated with miR-21–5 p

Since circRNAs are the main regulators for miRNAs activity and act as ceRNA to sponge miRNAs by complementary base pairing to regulate the expression of target mRNAs (*Li et al., 2018*; *Zheng et al., 2016*; *Cheng et al., 2019*), we analyzed the expression profile of circRNAs in endometrium samples from severe IUA patients and normal controls through high-throughput sequencing. We detected 17,134 distinct circRNAs and 4886 of them have been registered in circBase database (*Figure 2—figure supplement 1A*). We annotated these circRNAs and found that 4832 of them derived from protein-coding exons (*Figure 2—figure supplement 1B*). With the criteria of absolute mean fold change > 2.0 and p-values<0.05. The results showed that there were 263 upregulated circRNAs and 195 downregulated circRNAs (*Figure 2—figure supplement 1C*). Then, we predicted circRNAs with binding sites for miR-21–5 p based on miRanda database, and found that there were 172 candidate circRNAs (*Figure 2—figure supplement 1D* and *Supplementary file 3*). Of the 172 miR-21–5 p binding circRNAs, 12 were upregulated and 4 were downregulated in IUA patients (*Figure 2A,B*). We focused on the upregulated circRNA candidates because miR-21–5 p was downregulated in the endometria of IUA patients (*Figure 1C,D*) and circRNAs function as ceRNAs to inhibit miRNAs activities and expression. The most upregulated circRNA was hsa_circ_0003764 (circBase ID), and it was 64-fold higher than that in controls (*Figure 2B*). qRT-PCR (*Figure 2C*) and RNAscope assay (*Figure 2D*) showed that hsa_circ_0003764 was significantly upregulated in the endometria of IUA patients and mainly expressed in EECs. Through circPrimer software, we found that hsa_circ_0003764 is derived from 5 to 8 exons of tyrosine–protein phosphatase non-receptor type 12 (PTPN12) gene. Therefore, we termed it circPTPN12 in following experiments (*Figure 2E*).

We designed specific divergent primers to prove the existence of circPTPN12 in the endometria of IUA patients by PCR. The predicted spliced junction of circPTPN12 was validated with agarose gel electrophoresis (*Figure 2F*, top). The resultant products of divergent primers were confirmed in line with the sequence of circPTPN12 by sequencing (*Figure 2F*, bottom). To investigate the stability of circPTPN12, total RNA isolated from IUA patients' endometria was treated with RNase R for 15 min, then the expression of circPTPN12 and PTPN12 was respectively detected and the results showed that PTPN12 expression was significantly decreased while circPTPN12 expression was not changed (*Figure 2G*). Furthermore, we added actinomycin D to inhibit the transcription of EECs. We harvested total RNA at five time points, analyzed the expression of circPTPN12 and PTPN12 by qRT-PCR, and revealed that the half-life of circPTPN12 was exceeding 24 hr, whereas the half-life of linear PTPN12 transcript was less than 4 hr (*Figure 2H*). Thus, circPTPN12 was more stable than PTPN12 mRNA, suggesting that circPTPN12 is circular. Separation analysis of nuclear and cytoplasm showed that circPTPN12 was mainly distributed in the cytoplasm of EECs (*Figure 2I*).

Since miR-21–5 p was decreased and circPTPN12 was increased in fibrotic endometria of IUA patients, and both were expressed in EECs, we further analyzed the correlation between them. The results showed that circPTPN12 was negatively correlated with miR-21–5 p (*Figure 2J*).

## circPTPN12 functions as ceRNA to sponge miR-21–5 p

To prove the interaction between circPTPN12 and miR-21–5 p, we constructed a circPTPN12 plasmid to transfect HEK-293T cells. The transfection efficiency was verified by qRT-PCR (*Figure 3A*). We used biotin-labeled circPTPN12 to pull down the circPTPN12 in the lysates of the transfected HEK-293T cells. The electrophoresis showed that miR-21–5 p was pulled down by circPTPN12 (*Figure 3B*). To determine the binding site of circPTPN12 for miR-21–5 p, we cloned the full-length sequence of circPTPN12 in pGL3 vector containing a luciferase reporter. Luciferase activity was

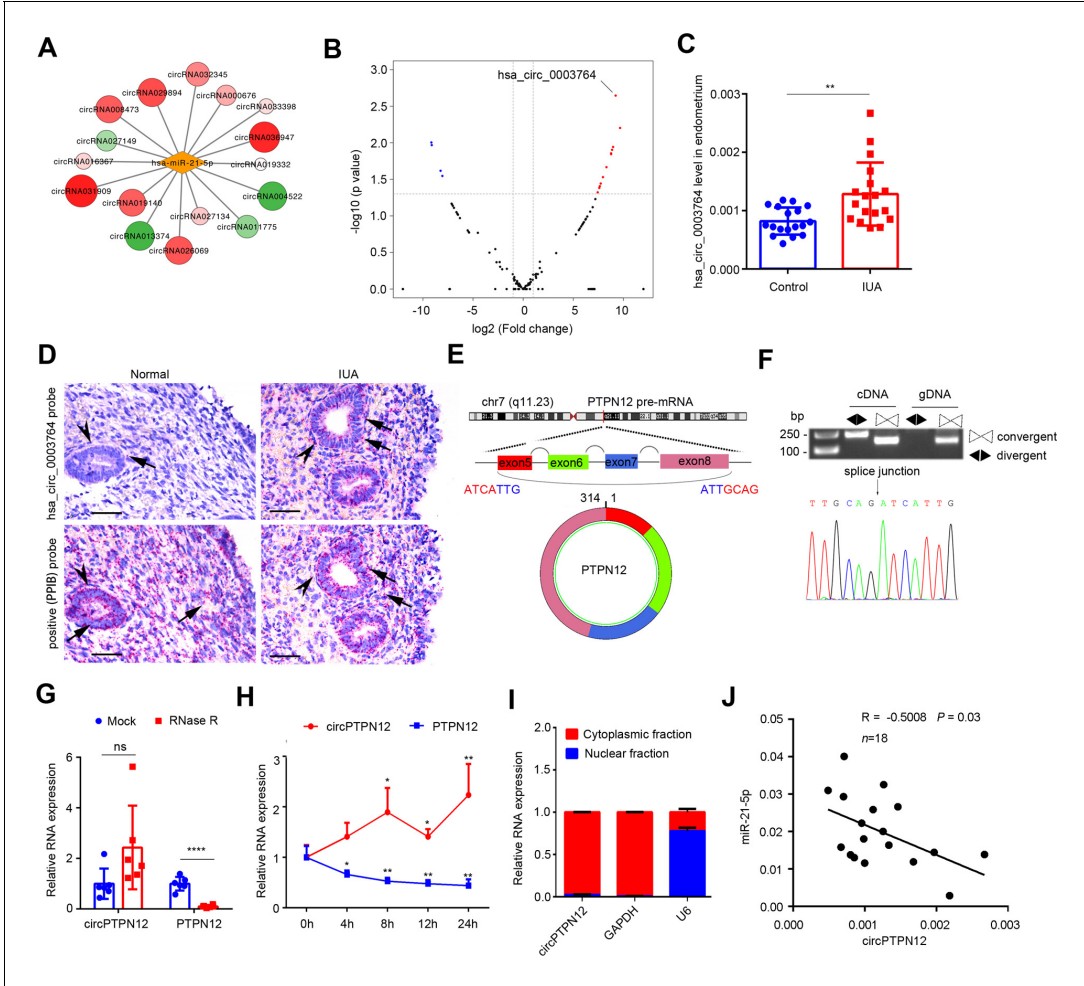

**Figure 2.** Upregulation of circPTPN12 in endometria of IUA patients is negatively correlated with miR-21–5 p. (**A**) Network diagram of circRNAs that have binding sites for miR-21–5 p and that significantly express in the endometria from severe IUA patients (n = 3) and controls (n = 3) based on the high-throughput RNA-sequencing analysis. (**B**) Volcano plots of circRNAs possessing binding sites for miR-21–5 p in the endometria from severe IUA patients (n = 3) and controls (n = 3) based on the high-throughput RNA-sequencing analysis. The red dots represent the circRNAs upregulated, and the blue dots represent the circRNAs downregulated (p<0.05 and fold change > 2). (**C**) Hsa_circ_0003764 expression levels in endometria of severe IUA patients (n = 18) and controls (n = 18) determined by qRT-PCR. (**D**) Representative images of RNA-scope assay using specific probes to detect hsa_circ_0003764 in the endometria of severe IUA patients (n = 3) and controls (n = 3). Peptidylprolyl isomerase B (PPIB) serves as the positive control. Arrowhead: epithelial cells; arrow (red dots): PPIB or has_circ_0003764 positive dots. Scale bars 50 μm. (**E**) Schematic illustration of the genomic loci of tyrosine-protein phosphatase non-receptor type 12 (PTPN12) gene and the cyclization of circPTPN12. (**F**) Verification of circPTPN12 in endometria. Top: Agarose gel electrophoresis shows that divergent primers amplified circPTPN12 in complementary DNA (cDNA) but not in genomic DNA (gDNA). Bottom: Sanger sequencing of the amplified band with divergent primers shows the spliced junction of circPTPN12. (**G**) qRT-PCR analysis of circPTPN12 and PTPN12 mRNA levels in total RNA extracted from endometria of severe IUA patients (n = 6) with or without RNase R treatment. (**H**) qRT-PCR analysis of circPTPN12 and PTPN12 mRNA levels in EECs (n = 4) treated with actinomycin D at the indicated time points. (**I**) circPTPN12 expression level in the nuclear and cytoplasmic fractions of EECs (n = 4) determined by qRT-PCR. (**J**) The correlation of circPTPN12 and miR-21–5 p in endometria of IUA patients (n = 18). *Spearman's* correlation coefficient R = –0.5008, p=0.03. (**C**), (**G**), and (**H**) Error bars indicate mean ± SD. No statistical difference (ns), *p<0.05, **p<0.01, ****p<0.0001.

The online version of this article includes the following source data and figure supplement(s) for figure 2:

**Source data 1.** qRT-PCR data for hsa_circ_0003764 relative expression.

**Source data 2.** Uncropped gels for *Figure 2F*.

**Source data 3.** qRT-PCR data for circPTPN12 and PTPN12 relative expression with or without RNase R treatment.

**Source data 4.** qRT-PCR data for circPTPN12 and PTPN12 relative expression with actinomycin D at the indicated time points.

**Source data 5.** qRT-PCR data for circPTPN12, GAPDH, and U6 relative expression in the nuclear and cytoplasmic fractions of EECs.

**Figure supplement 1.** circRNAs expression profile.

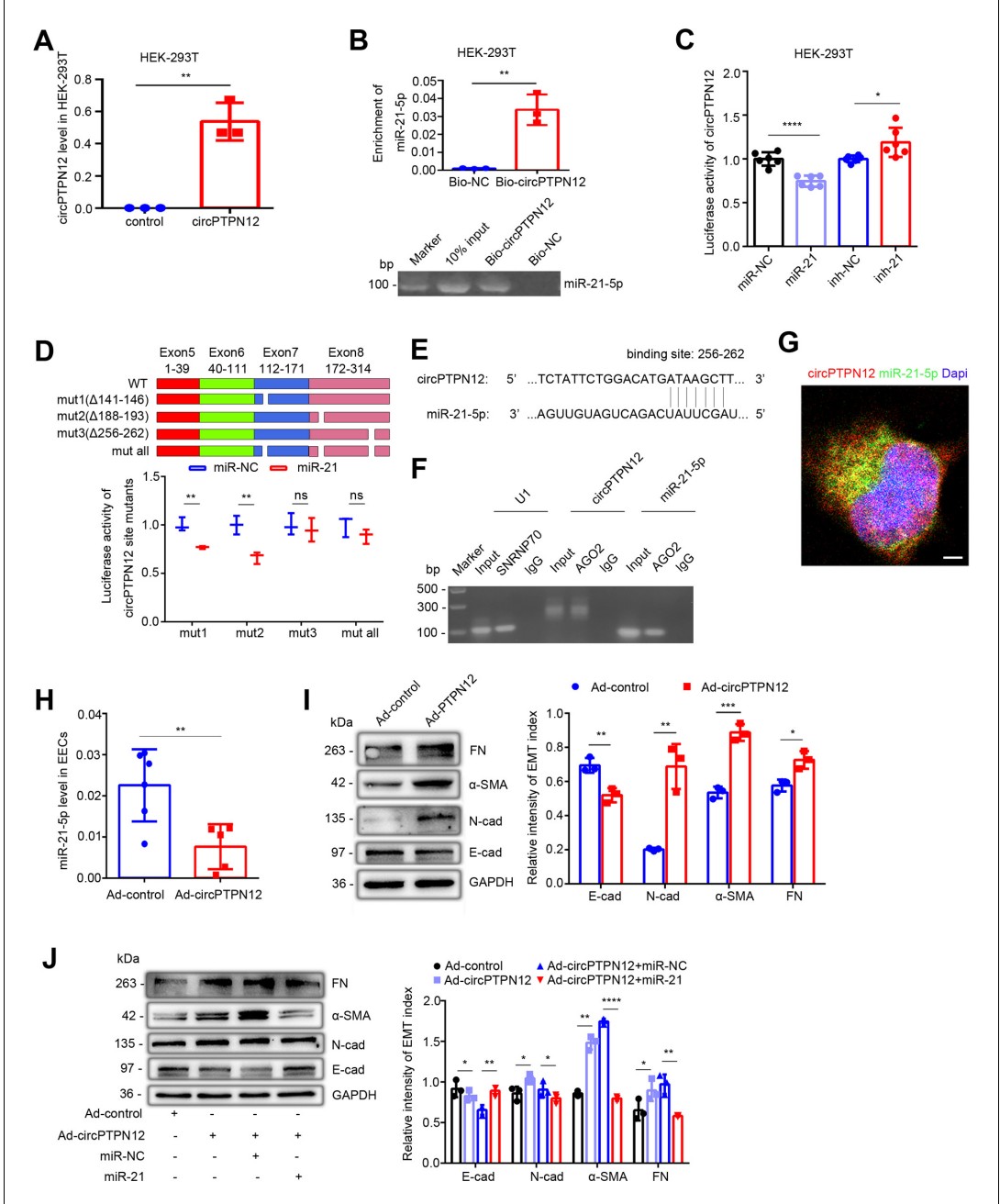

**Figure 3.** circPTPN12 functions as ceRNA to sponge miR-21–5 p. (**A**) qRT-PCR analysis of circPTPN12 expression level in circPTPN12 plasmid transfected (24 hr) HEK-293T cells (n = 3). (**B**) Capture of miR-21–5 p by circPTPN12. Top: qRT-PCR analysis of miR-21–5 p pulled down by biotin-labeled circPTPN12 (Bio-circPTPN12) or scramble (Bio-NC) probe from the HEK-293T cell lysates transfection with circPTPN12 plasmid (n = 3). Bottom: Amplified qRT-PCR products in agarose gel electrophoresis. (**C**) Luciferase activity of circPTPN12 in HEK-293T cells transfected with miR-21–5 p mimic (n = 6) or inhibitor (n = 6). (**D**) Top: Schematic diagram of three mutated sites in circPTPN12 luciferase reporter gene. Bottom: Luciferase activity of Luc-circPTPN12 containing single or all mutated sites of three putative miR-21–5 p binding sites in HEK-293T cells transfected with miR-21–5 p mimic (n = 3). (**E**) Simulated diagram of the exact binding sites between circPTPN12 and miR-21–5 p. (**F**) Enrichment of circPTPN12 and miR-21–5 p in AGO2 immunoprecipitates of HEK-293T cells transfected with circPTPN12 plasmid in agarose gel electrophoresis. (**G**) Colocalization between circPTPN12 and miR-21–5 p by RNA FISH in Ishikawa cells. Nuclei were stained with DAPI. Scale bar, 1 μm. (**H**) qRT-PCR analysis of miR-21–5 p level in adenovirus containing circPTPN12 (Ad-circPTPN12) or adenovirus containing no circPTPN12 (Ad-control) infected (48 hr) EECs (n = 5). (**I**) Left: FN, α-SMA, N-cad, E-cad, and GAPDH protein levels determined by western blotting in circPTPN12-infected (72 hr) EECs (n = 3). Right: The quantitative band intensities determined by image J software. (**J**) Left: FN, α-SMA, N-cad, E-cad, and GAPDH protein levels determined by western blotting in EECs transfected with miR-21–5 p mimic or miR-NC for 48 hr in the presence of Ad-circPTPN12 (n = 3). Right: The quantitative band intensities determined by image J software. (**A**) – (**D**) and (**H**) – (**J**) Error bars indicate mean ± SD. No statistical difference (ns), *p<0.05, **p<0.01, ***p<0.001, ****p<0.0001.

*Figure 3 continued on next page*

*Figure 3 continued*

The online version of this article includes the following source data and figure supplement(s) for figure 3:

**Source data 1.** qRT-PCR data for circPTPN12 relative expression.

**Source data 2.** qRT-PCR data for miR-21–5 p relative expression.

**Source data 3.** Data on luciferase activity in HEK-293T cells transfected with miR-21–5 p mimic or inhibitor.

**Source data 4.** Data on luciferase activity of Luc-circPTPN12 containing single or all mutated sites of three putative miR-21–5 p binding sites in HEK-293T cells transfected with miR-21–5 p mimic.

**Source data 5.** qRT-PCR data for miR-21–5 p relative expression in EECs with Ad-control or Ad-circPTPN12 treatment.

**Source data 6.** Uncropped western blots for *Figure 3I*.

**Source data 7.** The quantitative band intensities for *Figure 3I*.

**Source data 8.** Uncropped western blots for *Figure 3J*.

**Source data 9.** The quantitative band intensities for *Figure 3J*.

**Figure supplement 1.** circPTPN12 adenovirus (Ad-circPTPN12) construction.

---

substantially reduced in the presence of the miR-21–5 p mimic, which was effectively restored when miR-21–5 p was knockdown (*Figure 3C*). We then compared the sequence of circPTPN12 with that of miR-21–5 p using circBank and Circular RNA Interactome databases and noticed that circPTPN12 contains three putative target sites of miR-21–5 p (*Figure 3D*, top). The test of delete mutations of each or all of three putative binding sites showed that only the third site was validated to have the capacity of binding by the luciferase assay (*Figure 3D*, bottom). The schematic diagram of real binding site is shown in *Figure 3E*. To further verify the binding of circPTPN12 with miR-21–5 p, we conducted RNA immunoprecipitation (RIP) experiment to precipitate the complex of argonaute (AGO) protein based on a principle that miRNAs exert post-transcriptional regulation through binding with AGO protein to form the targeting module of the miRNA-induced silencing complex (miRISC) (*Gebert and MacRae, 2019*). The RIP experiment showed that the precipitates of HEK-293T cell lysates stably expressing Flag-AGO2 or Flag-GFP contained not only miR-21–5 p, but also circPTPN12 (*Figure 3F*), indicating the bind of circPTPN12 and miR-21–5 p. Fluorescence in situ hybridization (FISH) demonstrated that miR-21–5 p was colocalized with circPTPN12 in the cytoplasm of Ishikawa cell (*Figure 3G*).

To investigate the effect of circPTPN12 on the function of miR-21–5 p, we constructed recombinant adenovirus harboring exons 5–8 of PTPN12 (Ad-circPTPN12) along with approximately 1 kb flanking intron sequences containing complementary Alu elements (*Figure 3—figure supplement 1A*). The expression of circPTPN12 increased seven times in EECs after Ad-circPTPN12 infection (*Figure 3—figure supplement 1B*). Importantly, qRT-PCR confirmed that circPTPN12 overexpression decreased the miR-21–5 p level in primary EECs (*Figure 3H*). Meanwhile, compared with that of adenovirus containing no circPTPN12 (Ad-control), overexpression of circPTPN12 enhanced protein abundance of N-cadherin, α-SMA, and FN, but decreased the level of E-cadherin in EECs (*Figure 3I*). While re-transfected with miR-21–5 p mimic reversed this phenomenon with upregulated E-cadherin expression and downregulated N-cadherin, α-SMA, and FN expression (*Figure 3J*).

## Downregulation of miR-21–5 p by circPTPN12 promotes EEC–EMT through upregulating ΔNp63α

Since circPTPN12 inhibited miR-21–5 p, activity to promote EEC–EMT and miR-21–5 p may be the antagonizer of ΔNp63α that was predicted by Microrna.org database. We conducted the luciferase reporter assay with the full-length 3′-UTR of ΔNp63α$^{wild}$ and ΔNp63α$^{mut}$ targeted by miR-21–5 p (*Figure 4A*). In line with the prediction, transfection with miR-21–5 p mimic significantly reduced luciferase activity in HEK-293T cells transfected with ΔNp63α$^{wild}$ plasmid, whereas knockout of miR-21–5 p in ΔNp63α$^{wild}$ HEK-293T cells enhanced the luciferase activity; however, these effects were not observed in HEK-293T cells transfected with ΔNp63α$^{mut}$ plasmid (*Figure 4B*). We used primary EECs transfected with the miR-21-mimic or inhibitor for 48 hr following infection with recombinant adenovirus containing ΔNp63α (Ad-ΔNp63α) and empty vector (Ad-CTL), respectively, and found that miR-21–5 p mimic remarkably downregulated the mRNA and protein of ΔNp63α in ΔNp63α + EECs (*Figure 4C,D*). And the miR-21–5 p inhibitor upregulated ΔNp63α protein level in ΔNp63α + EECs (*Figure 4E*). Meanwhile, upregulation of circPTPN12 increased luciferase activity in HEK-293T cells transfected with ΔNp63α$^{wild}$ plasmids, while no luciferase activity change was

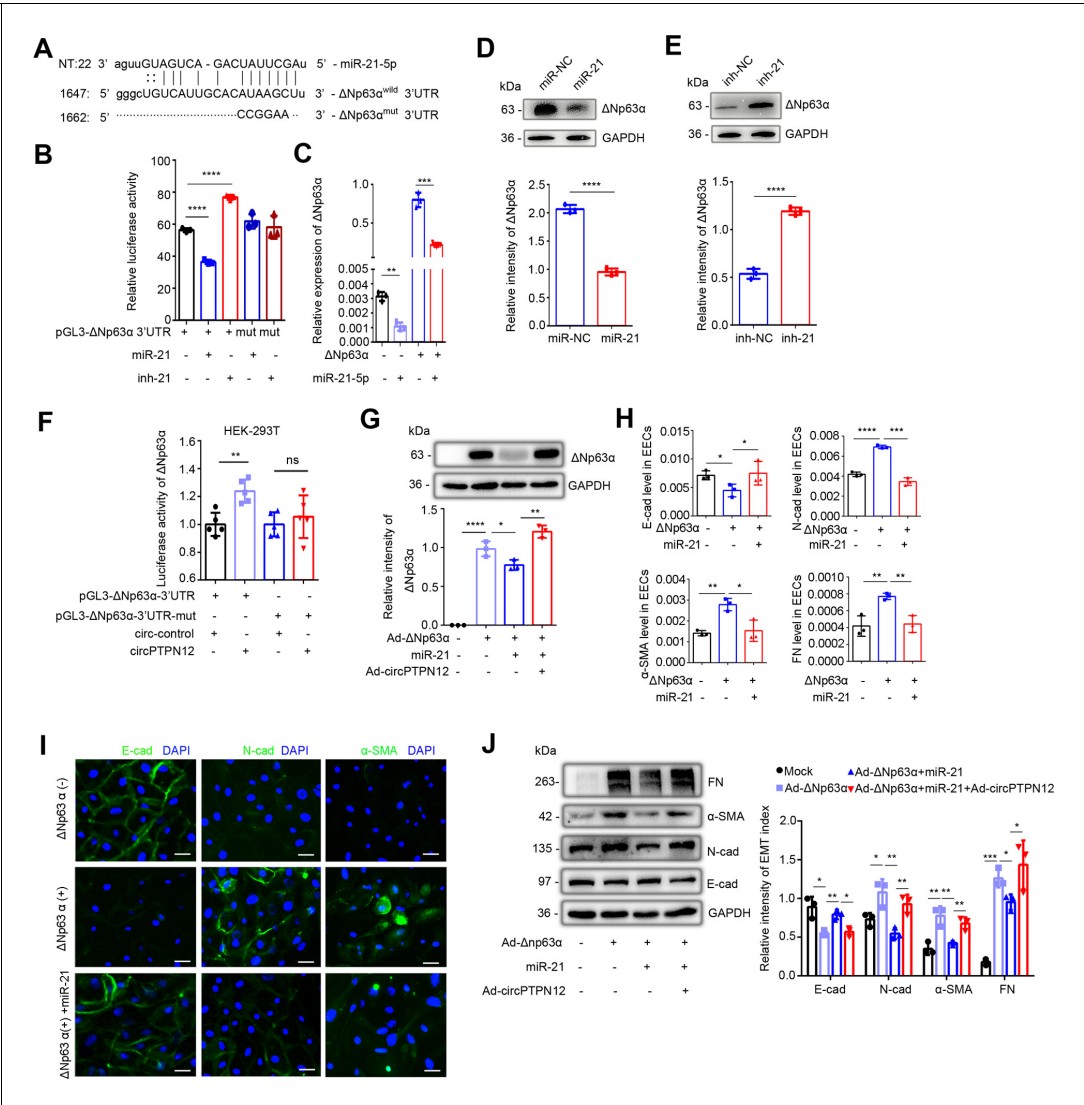

**Figure 4.** circPTPN12-downregulated miR-21–5 p promotes EEC–EMT through upregulation of ΔNp63α. (**A**) The putative site for the interaction between miR-21–5 p and the 3′-UTR of ΔNp63α. The predicted miR-21–5 p binding sequence AUAAGC was replaced by CCGGAA for pGL3-Δ Np63α$^{mut}$ construction. (**B**) Luciferase activity of ΔNp63α$^{wild}$ and ΔNp63α$^{mut}$ in HEK-293T cells (n = 3) co-transfected with miR-21–5 p mimic or inhibitor for 24 hr. (**C**) qRT-PCR analysis of ΔNp63α levels in adenovirus containing ΔNp63α (Ad-ΔNp63α) infected EECs transfected with miR-21–5 p mimic (n = 3). (**D**) Top: ΔNp63α protein expression in miR-21–5 p mimic transfected-ΔNp63α + EECs (48 hr) determined by western blotting (n = 3). Bottom: The quantitative band intensities determined by image J software. (**E**) Top: ΔNp63α protein expression in miR-21–5 p inhibitor transfected-ΔNp63α + EECs (48 hr) determined by western blotting (n = 3). Bottom: The quantitative band intensities determined by image J software. (**F**) Luciferase activity of ΔNp63α$^{wild}$ and ΔNp63α$^{mut}$ in HEK-293T cells co-transfected with circPTPN12 or circ-control plasmids (n = 5). (**G**) Top: ΔNp63α protein expression determined by western blotting in Ad-circPTPN12-infected ΔNp63α+ or ΔNp63α− EECs in the presence of miR-21–5 p mimic or miR-NC for 48 hr (n = 3). Bottom: The quantitative band intensities determined by image J software. (**H**) qRT-PCR analysis of E-cad, N-cad, α-SMA, and FN mRNA levels in miR-21–5 p mimic or miR-NC transfected ΔNp63α+ EECs (n = 3). (**I**) Representative images (n = 3) of E-cad, N-cad, and α-SMA immunofluorescence staining in miR-21–5 p mimic or miR-NC transfected ΔNp63α + EECs. Scale bars, 20 μm. (**J**) Left: FN, α-SMA, N-cad, E-cad, and GAPDH protein levels determined by western blotting in Ad-circPTPN12-infected ΔNp63α+ or ΔNp63α− EECs in the presence of miR-21–5 p mimic or miR-NC for 48 hr (n = 3). Right: The quantitative band intensities determined by image J software. (**B**) – (**H**) and (**J**) Error bars indicate mean ± SD. *p<0.05, **p<0.01, ***p<0.001, ****p<0.0001.

The online version of this article includes the following source data for figure 4:

**Source data 1.** Data on luciferase activity of Luc-ΔNp63α wild or mut in HEK-293T cells transfected with miR-21–5 p mimic or inhibitor.

**Source data 2.** qRT-PCR data for ΔNp63α relative expression in EECs transfected with miR-21–5 p mimic.

**Source data 3.** Uncropped western blots for *Figure 4D,E*.

**Source data 4.** The quantitative band intensities for *Figure 4D*.

**Source data 5.** The quantitative band intensities for *Figure 4E*.

**Source data 6.** Data on luciferase activity of Luc-ΔNp63α wild or mut in HEK-293T cells transfected with circ-control or circPTPN12.

**Source data 7.** Uncropped western blots for *Figure 4G*.
**Source data 8.** The quantitative band intensities for *Figure 4G*.
**Source data 9.** qRT-PCR data for E-cad, N-cad, α-SMA, and FN relative expression.
**Source data 10.** Uncropped western blots for *Figure 4J*.
**Source data 11.** The quantitative band intensities for *Figure 4J*.

observed when HEK-293T transfected with ΔNp63α$^{mut}$ plasmids (*Figure 4F*). Moreover, the upregulation of circPTPN12 counteracted the inhibitory effect of miR-21–5 p on ΔNp63α expression (*Figure 4G*). And that we revealed that miR-21–5 p mimic remitted ΔNp63α-induced EEC–EMT, upregulated E-cadherin and downregulated N-cadherin and α-SMA at both the mRNA and protein levels (*Figure 4H,I*). Furthermore, upregulation of circPTPN12 counteracted the reversal effect of miR-21–5 p on ΔNp63α-induced EMT in EECs (*Figure 4J*).

## miR-21–5 p alleviates circPTPN12-induced EEC–EMT in IUA-like mouse model

We developed a mouse model by uterine mechanical injury to simulate IUA in human as described previously (*Zhao et al., 2020*; *Yang et al., 2017*). Compared with sham-operated mice, the mechanically injured mice had mildly upregulated *N-cadherin* and *α-SMA* in the luminal epithelial cells of endometria, slightly downregulated *E-cadherin* in luminal and glandular EECs, and minimally changed in Masson staining (*Figure 5—figure supplement 1A*).

Based on circBase and circBank database, mouse does not express circPTPN12, which provided us an opportunity to test the role of circPTPN12 in the pathogenesis of endometrium fibrosis by intrauterine injection of a circPTPN12-containing recombinant adeno-associated virus with a green label (AAV-circPTPN12). The flowchart of intrauterine injection is presented in *Figure 5—figure supplement 1B*. The green fluorescence in mouse endometria was clearly observed after four estrous cycles following virus injection (*Figure 5—figure supplement 1C*), indicating that AAV-circPTPN12 infected the mouse endometrium. We further validated the increased circPTPN12 levels in the endometria by qRT-PCR (*Figure 5A*). With the increase of circPTPN12 expression, the expression of *miR-21–5* p decreased significantly (*Figure 5B*). Furthermore, the mice injected with AAV-circPTPN12 showed increased mRNA levels of *N-cadherin*, *α-SMA*, and *FN* and decreased mRNA level of *E-cadherin*, compared with those injected with adeno-associated virus with empty vector (*Figure 5C*). Immunohistochemical staining displayed that AAV-circPTPN12 injection clearly reduced *E-cadherin* expression in EECs and remarkably upregulated *N-cadherin* and *α-SMA* in both epithelial and stroma cells (*Figure 5D*). In addition, Masson staining showed strong positive (*Figure 5D*). The results of EMT and severe endometrial fibrosis in mice caused by AAV-circPTPN12 intrauterine injection were similar to the results induced by dual-injury, namely mechanical injury with lipopolysaccharide injection (*Zhao et al., 2020*). *ΔNp63α* mRNA was also significantly upregulated after AAV-circPTPN12 administration (*Figure 5E*).

Since circPTPN12 served as the ceRNA of miR-21–5 p and the pro-EMT effect of circPTPN12 was reversed by miR-21–5 p in vitro (*Figure 3B–I*), we further conducted the mouse experiment by injecting agomir-21–5 p or agomir negative control (agomir-NC) into the uterine cavity of the AAV-circPTPN12 mouse model every 5 days for three times (*Figure 5—figure supplement 1D,E*). The results showed that *miR-21–5* p in the endometria was overexpressed (*Figure 5F*) and the mRNA level of *E-cadherin* was elevated, while the mRNA levels of *N-cadherin*, *α-SMA*, and *FN* were decreased (*Figure 5G*). Immunohistochemical results showed that agomir-21–5 p application significantly increased *E-cadherin* and reduced *N-cadherin* and *α-SMA* in both luminal and glandular epithelia and reduced Masson staining compared with agomir-NC injection (*Figure 5H*).

## Discussion

In the present study, we revealed that in normal endometrium, the high level of miR-21–5 p in EECs inhibits ΔNp63α expression to maintain the homeostasis of EECs, and in the pathological circumstance, circPTPN12 is highly expressed, which decreases miR-21–5 p level in EECs, so the suppression effect of miR-21–5 p on the expression of ΔNp63α is counteracted, leading to EEC–EMT and

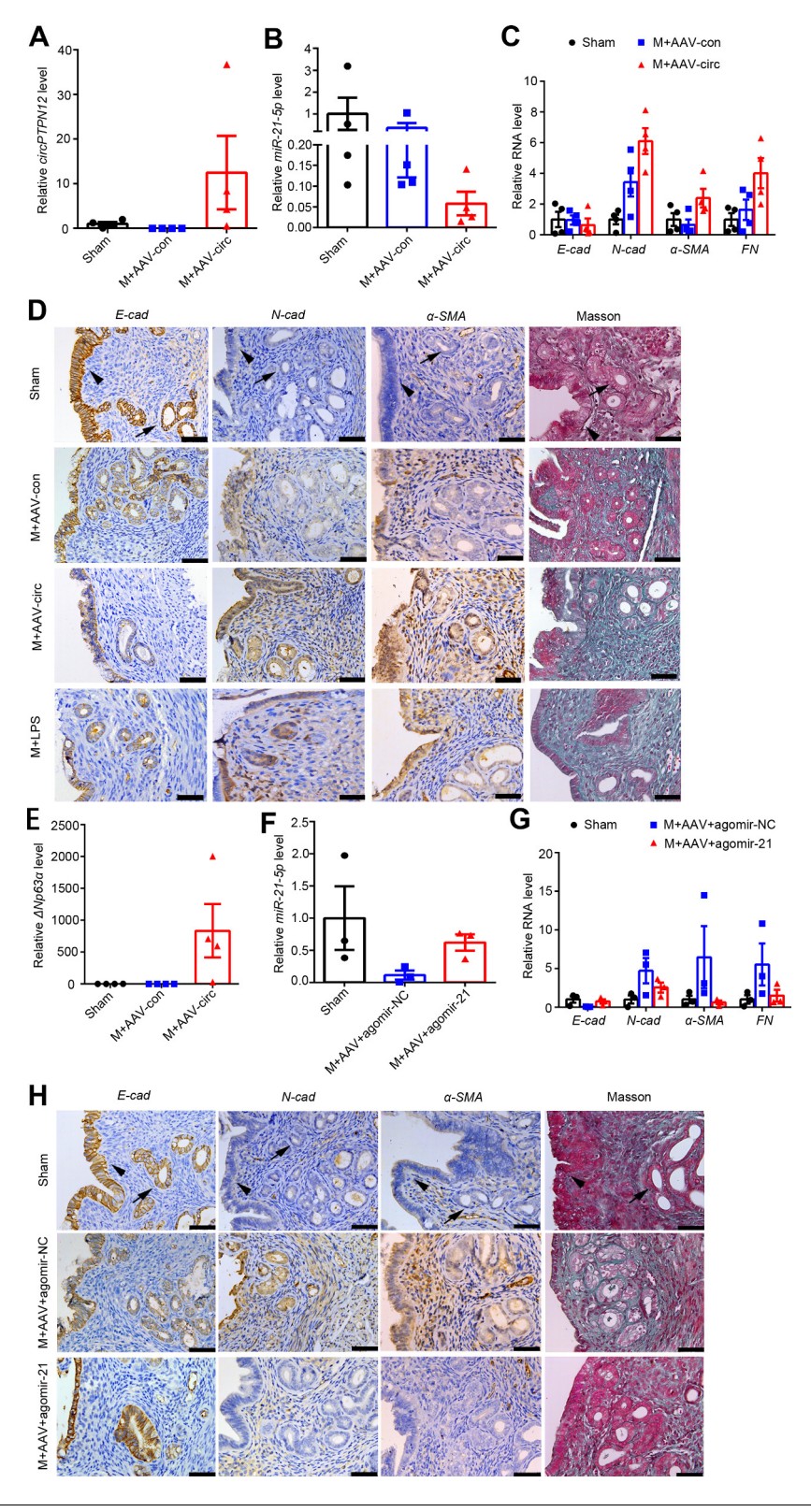

**Figure 5.** Intrauterine injection of miR-21–5 p alleviates circPTPN12-induced EEC–EMT in vivo. (A) – (E) qRT-PCR analysis of *circPTPN12* levels (A), *miR-21–5* p levels (B), *E-cad*, *N-cad*, *α-SMA*, and *FN* mRNA levels (C), representative immunohistochemistry images of *E-cad*, *N-cad*, *α-SMA*, and Masson staining (D, the first three rows), *ΔNp63α* levels (E) in endometria of mice with sham-operation (Sham, n = 4), mechanical injury and adeno-associated virus (AAV)-control injection (M + AAV con, n = 4), mechanical injury and AAV-circPTPN12 injection (M + AAV circ, *n* = 4). (D, the fourth row)

*Figure 5 continued on next page*

*Figure 5 continued*

Representative immunohistochemistry images of *E-cad*, *N-cad*, *α-SMA*, and Masson staining in endometria of mice with mechanical injury and lipopolysaccharide injection (M + LPS, n = 4). (F) – (H) qRT-PCR analysis of *miR-21–5* p levels (F), *E-cad*, *N-cad*, *α-SMA*, and *FN* mRNA levels (G), representative immunohistochemistry images of *E-cad*, *N-cad*, *α-SMA*, and Masson staining (H) in endometria of mice with sham-operation (n = 3), mechanical injury with AAV-circPTPN12 and agomir-NC injection (M + AAV + agomiR-NC, n = 3), mechanical injury with AAV-circPTPN12 and agomir-21–5 p injection (M + AAV + agomiR-21, n = 3). Scale bars, 50 μm. Arrow head: luminal epithelial cells; arrow: glandular epithelial cells. (A) – (C), (E) – (F), and (G) Error bars indicate mean ± SEM.

The online version of this article includes the following source data and figure supplement(s) for figure 5:

**Source data 1.** qRT-PCR data of *circPTPN12* relative expression for *Figure 5A*.
**Source data 2.** qRT-PCR data of *miR-21–5* p relative expression for *Figure 5B*.
**Source data 3.** qRT-PCR data of *E-cad*, *N-cad*, *α-SMA*, and *FN* relative expression for *Figure 5C*.
**Source data 4.** qRT-PCR data of *ΔNp63α* relative expression for *Figure 5E*.
**Source data 5.** qRT-PCR data of *miR-21–5* p relative expression for *Figure 5F*.
**Source data 6.** qRT-PCR data of *E-cad*, *N-cad*, *α-SMA*, and *FN* relative expression for *Figure 5G*.
**Figure supplement 1.** IUA-like mouse model construction.

endometrial fibrosis. Therefore, we propose a working model to explain the role of circPTPN12/miR-21–5 p/ΔNp63α pathway in the pathogenesis of human endometrial fibrosis (*Figure 6*).

Previously, we observed that ΔNp63α is ectopically expressed in EECs in IUA patients (*Zhao et al., 2017*), and showed that ΔNp63α promotes the expression of SNAI1 by DUSP4/GSK3B pathway and induces EEC–EMT and endometrial fibrosis (*Zhao et al., 2020*). These findings are in agreement with the therapeutic effect of transplantation of mesenchymal stem cells into the uterine cavity (*Cao et al., 2018*). Since miRNAs are main regulator of ΔNp63α (*Candi et al., 2015*; *Lena et al., 2008*; *Rodriguez Calleja et al., 2016*) and sufficient abundance of miRNAs transcripts in cells is essential in the regulating target genes (*Liu et al., 2019a*; *Brown and Naldini, 2009*), we chose the highest expressed miR-21–5 p in normal endometria to study its role in the current study. We found that miR-21–5 p is located in EECs, and the downregulation of miR-21–5 p can increase the expression of ΔNp63α, leading to the transition of EECs to cells with features of mesenchymal cells. In IUA mice model, supplementation of exogenous miR-21–5 p reversed the EEC–EMT and endometrial fibrosis, indicating that miR-21–5 p is essential in maintaining EECs properties. Our

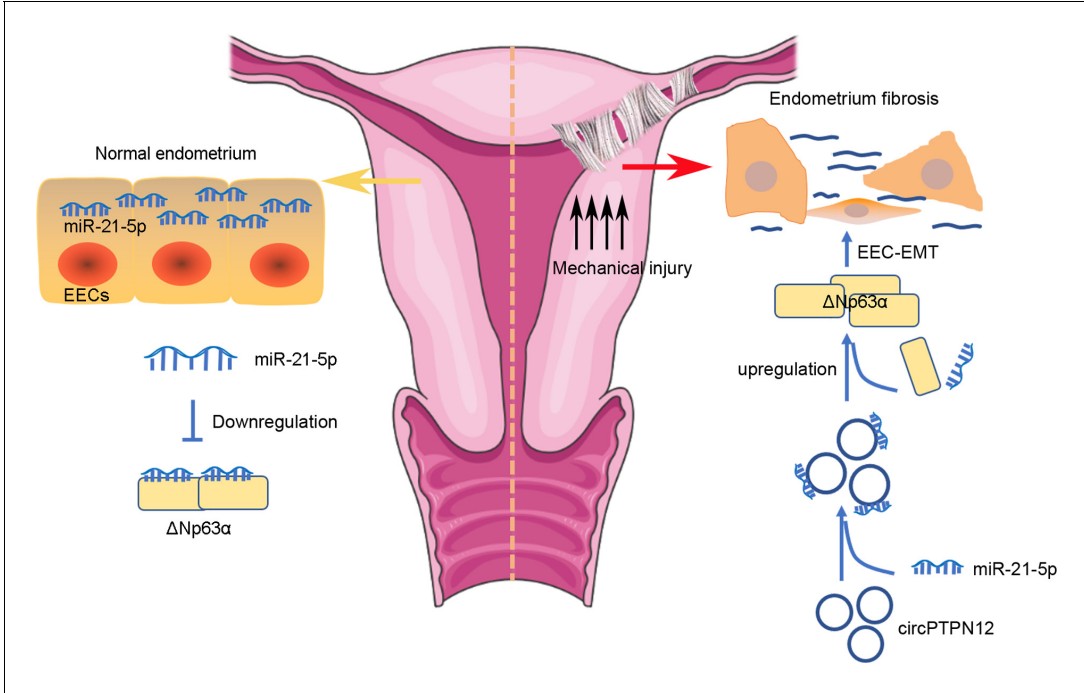

**Figure 6.** Proposed pathogenesis of endometrium fibrosis.

findings appear to be different from the observations that in lung and kidney fibrosis researches miR-21–5 p promotes EMT (*Liu et al., 2019b*; *Wang et al., 2019*; *Luo et al., 2019*; *Liu et al., 2019c*). Thus, we consider that miR-21–5 p may play different roles in different tissues, which merits further study.

CircRNAs are important regulators of miRNAs and play critical roles in the pathogenesis of lung/renal/liver fibrosis. In this study, we identified a new circPTPN12, which rarely expressed in normal endometrium but significantly upregulated in the endometria of IUA patients. The colocalization of circPTPN12 and miR-21–5P in the cytoplasm of EECs, luciferase assay, and RIP experiments supports the circPTPN12/miR-21–5 p sponge hypothesis. Therefore, the current study for the first time demonstrates the role of a circRNA in endometrial fibrosis.

There are some limitations in our study. First, since the IUA patients enrolled in this study were all associated with repeated curettage, the findings in the present study may not be applicable to the IUA patients associated with infection or uterine artery embolization. Second, we only studied the roles of miR-21–5 p and circPTPN12 based on their highest expressions in the heatmap or volcano plots, yet there may be other less-expressed miRNAs and circRNAs that are potentially associated with the endometrial fibrosis, which remains to be further studied.

# Materials and methods

**Key resources table**

| Reagent type (species) or resource | Designation | Source or reference | Identifiers | Additional information |
|---|---|---|---|---|
| Cell line (*H. sapiens*) | HEK-293T | ATCC | RRID:CVCL_0063 | |
| Cell line (*H. sapiens*) | Ishikawa | ATCC | RRID:CVCL_2529 | |
| Antibody | Anti-E-cadherin (mouse monoclonal) | Abcam | Cat# ab76055 | WB (1:1000) IHC (1:400) |
| Antibody | Anti-N-cadherin (rabbit monoclonal) | Abcam | Cat# ab18203 | WB (1:500) IHC (1:400) |
| Antibody | Anti-α-SMA (rabbit monoclonal) | Abcam | Cat# ab5694 | WB (1:500) IHC (1:100) |
| Antibody | Anti-FN (rabbit monoclonal) | Abcam | Cat# ab2413 | WB (1:500) |
| Antibody | Anti-ΔNp63α (rabbit monoclonal) | Millipore | Cat# ABS552 | WB (1:500) |
| Antibody | Anti-AGO2 (rabbit monoclonal) | Abcam | Cat# ab186733 | RIP (1:1000) |
| Antibody | Anti-GAPDH (HRP-conjugated GAPDH mouse mAb) | ABclonal | Cat# AC035 | WB (1:10,000) |
| Antibody | Anti-rabbit IgG | Cell Signaling Technology | Cat# 7074S | WB (1:2000) |
| Antibody | Anti-mouse IgG | Cell Signaling Technology | Cat# 7076S | WB (1:2000) |
| Sequenced based reagent | RT-qPCR primers | This paper | | see *Supplementary file 5* |
| Commercial assay or kit | NE-PER Nuclear and Cytoplasmic Extraction Reagents | Thermo Scientific | Cat# 78835 | |
| Commercial assay or kit | PrimeScript reagent Kit with DNA Eraser | TaKaRa BIO | Cat# RR047B | |
| Commercial assay or kit | RNA-scope assay | Advanced Cell Diagnostics | Cat# 710171 Cat# 713661 Cat# 854881 | |
| Chemical compound, drug | Actinomycin D | MedChem Express | Cat# HY-17559 | |
| Software, algorithm | Prism version 6.0 | GraphPad Software Inc | RRID:SCR_002798 | |

*Continued on next page*

*Continued*

| Reagent type (species) or resource | Designation | Source or reference | Identifiers | Additional information |
|---|---|---|---|---|
| Software, algorithm | Image J | NIH | RRID:SCR_003070 | |
| Software, algorithm | edgeR | Bioconductor | RRID:SCR_012802 | |

### Human endometrium samples

This study was approved by the Committee on Human Research of the Nanjing Drum Tower Hospital (No. 2012022), and informed consent was obtained from each participant. Human endometrium samples were collected in the late proliferative phase of the menstrual cycle from child-bearing age women during hysteroscopy for infertility screening at the Affiliated Drum Tower Hospital of Nanjing University from January 2014 to December 2016. The late proliferative phase was defined based on follicle size between 15 and 18 mm by ultrasonography and a low level of serum progesterone. This study enrolled 21 normal controls, who were infertile patients with normal ovary function, normal uterine cavity, and endometrium thickness $\geq$ 8 mm immediate before ovulation. Twenty-one severe IUA patients were diagnosed as score > 8 based on criteria recommended by the American Fertility Society (*The American Fertility Society classifications of adnexal adhesions, 1988*). The clinical information of all patients and controls was listed in *Supplementary file 4*. Fibrotic endometrial samples were taken from two sites in the uterine body or fundus that showed the most severe adhesion. Normal control samples were taken in the uterine body and fundus. Half of each sample was stored in liquid nitrogen, and the other half was fixed with formalin and used for further experiments as described below.

### RNA-sequencing, bioinformatics analysis for miRNAs and circRNAs

Total RNA extracted from the endometrial samples was subjected to library construction, followed by sequencing on an Illumina Hiseq 250020 (miRNA) or Illumina Hiseq X10 platform (circRNA) by Vazyme (Nanjing, China). The raw reads were filtered to achieve the clean tags through in-house Perl processes to map to the reference genome. miRNAs were identified by mapping to miRBase database and circRNAs were identified by circRNA Finder database (*Fu and Liu, 2014*). The miRNAs and circRNAs expression level were calculated and normalized to transcripts per million (TPM), TPM = $10^6$ C/L, where C is the counts of a miRNA or circRNA, and L is counts sum of all miRNAs or circRNAs. To identify differentially expressed miRNAs and circRNAs between samples, the edgeR package (http://www.r-project.org/) was used. Differentially expressed miRNAs and circRNAs were identified with fold change > 2 or < −2 and p<0.05.

### Cell isolation and culture

The isolation and culture of EECs were described previously (*Zhao et al., 2017*). Endometrial tissues were treated with collagenase type I (Sigma, St Louis, MO), hyaluronidase (Sigma), and DNase (Roche, Indianapolis, IN), followed by filtration through a 40 μm cell strainer (BD Biosciences, San Jose, CA) to remove the stromal cells and then through a 100 μm cell strainer (BD Biosciences) to collect the EECs. The cells were plated on matrigel-coated dishes and cultured with keratinocyte serum-free medium (Gibco, Waltham, MA) containing 2% fetal bovine serum (FBS), 100 U/ml penicillin, and 0.1 mg/ml streptomycin. HEK-293T and Ishikawa cells were grown in Dulbecco's modified Eagle's medium containing 10% FBS, 100 U/ml penicillin, and 0.1 mg/ml streptomycin at 37°C with 5% $CO_2$. They are purchased from ATCC and authenticated using STR profiling by ATCC, and they are tested to be free from mycoplasma contamination.

### RNA and qRT-PCR

Total RNA was extracted by Trizol reagent (Invitrogen Life Technologies, Carlsbad, CA). For acquisition of RNA from cytoplasm and nucleus, respectively, the nuclear and cytoplasmic fractions were isolated using NE-PER Nuclear and Cytoplasmic Extraction Reagents (Thermo Scientific, Waltham, MA). The circRNAs were reverse-transcribed using PrimeScript reagent Kit with DNA Eraser (TaKaRa BIO, Japan). qRT-PCR was performed using SYBR qPCR Master Mix (Vazyme). U6 and GAPDH were

used as internal controls. The relative RNA levels were determined by the $2^{-\Delta Ct}$ or $2^{-\Delta\Delta Ct}$ method. All primers are listed in *Supplementary file 5*.

## Western blotting

After the determination of protein concentration of the cell lysates, each sample with equal amount of protein was subjected to SDS–PAGE (10% or 15%) and transferred to a PVDF membrane (Millipore, Burlington, MA). The membrane was incubated with the specific primary antibody overnight at 4˚C, followed by incubation with HRP-conjugated anti-rabbit IgG (1:2000, Cell Signaling Technology, Boston, MA, cat# 7074S) or anti-mouse IgG (1:2000, Cell Signaling Technology, cat# 7076S). The signals were visualized with ECL solution (Millipore). The primary antibodies were as follows: E-cad (1:1000, Abcam, Cambridge, UK, cat# ab76055), N-cad (1:500, Abcam, cat# ab18203), α-SMA (1:500, Abcam, cat# ab5694), FN (1:500, Abcam, cat# ab2413), and ΔNp63α (1:500, Millipore, cat# ABS552). The antibody for GAPDH (1:10,000, ABclonal, MA, cat# AC035) was used as a control.

## Immunohistochemistry and immunofluorescence

For immunohistochemistry, mice endometrium tissues were fixed in 10% formaldehyde for 8 hr at 4˚C, transferred to a tissue processor (Leica ASP300 S, Wetzlar, Germany) for dehydration and wax infiltration, and embedded in paraffin. Paraffin blocks were cut into 2 µm thick slices. After dewaxing and blocking endogenous peroxidase activity in 3% $H_2O_2$, slices were heat-mediated in universal antigen retrieval. The slides were blocked in 2% bovine serum albumin/PBST for 1 hr. After these pretreatments, slides were incubated with primary antibodies at 4˚C overnight and washed by PBST. HRP-conjugated secondary antibodies were incubated for 1 hr. Slides were exposed to 3′3-diaminobenzidine to visualize the antigen signals. The positive staining was confirmed in a blinded manner by two independent observers by microscopy (DMi8, Leica, Wetzlar, Germany). For cell immunofluorescence, isolated EECs that had been plated on matrigel-coated coverslips for varying times were fixed in 4% paraformaldehyde for 10 min and permeabilized with cold methanol. Fixed cells were stained with a primary antibody overnight, washed, and incubated with secondary antibodies conjugated to fluorescein or rhodamine. Nuclei of EECs were stained using 4′,6-diamidino-2-phenylindole (DAPI, Abcam). The primary antibodies were as follows: E-cad (1:400, Abcam), N-cad (1:400, Abcam), α-SMA (1:100, Abcam), and ΔNp63α (1:100, Millipore).

## RNA-scope assay

RNA-scope assay was performed on 5 µm thick tissue sections (Advanced Cell Diagnostics, San Francisco, CA). Sections were hybridized with peptidylprolyl isomerase B (cat# 710171), pre-miR-21 (cat# 713661), or circPTPN12 (cat# 854881) probes, respectively. Slides were viewed under a microscope (DMi8).

## Dual-luciferase reporter assay

The full-length sequence of circPTPN12 or the mutants were inserted into pGL3 luciferase vector to obtain the circPTPN12^wild or circPTPN12^mut constructs (Generay Biotechnology, Shanghai, China). The mutants were those with deletion of each or all the three sites of circPTPN12 for binding miR-21–5 p. 3′-UTR or the 3′-UTR with miR-21–5 p binding site mutation of ΔNp63α were inserted into a pGL3 plasmid to obtain the pGL3-ΔNp63α^wild or pGL3-ΔNp63α^mut constructs (Generay). HEK-293T cells were co-transfected with circPTPN12^wild or circPTPN12^mut or pGL3-ΔNp63α^wild or pGL3-ΔNp63α^mut and miR-21–5 p mimic (Ribobio, Guangzhou, China) or miR-21–5 p inhibitor (Ribobio) or circPTPN12 plasmid using Lipofectamine 3000. After 24 hr, the firefly and Renilla luciferase activities were measured consecutively using a Dual Luciferase Reporter Assay kit (Vazyme).

## Recombinant δNp63α adenovirus construction

ΔNp63α adenovirus (Ad-ΔNp63α) were constructed as previous study (*Zhao et al., 2017*). The open reading frame (GenBank: AF075431.1) and 3′-UTR of *Homo sapiens* of ΔNp63α were inserted into a pDC315-3FLAG-SV40-EGFP vector and ligated into a shuttle plasmid. Then, HEK-293A cells were co-transfected with the shuttle plasmid and adenoviral backbone plasmid to produce ΔNp63α recombinant adenoviral vector. The same vector was used without the ΔNp63α insertion to construct the control virus (GeneChem, Shanghai, China). circPTPN12 adenovirus (Ad-circPTPN12), adeno-

associated virus (AAV-circPTPN12), and plasmid construction circPTPN12 adenovirus were obtained from Genechem. Briefly, the full length of circPTPN12 with its flanking introns including complementary Alu elements was amplified to insert into a pCMV-MCS-EGFP vector, and an empty pCMV-MCS-EGFP vector served as control virus. circPTPN12 adeno-associated virus was obtained from HanBio Biotechnology (Shanghai, China). The full length of circPTPN12 was inserted into pHBAAV-CMV-CircRNA-EF1-ZsGreen vector and then inserted into adeno-associated virus. The empty vector was used as control virus. circPTPN12 overexpression plasmid was obtained from Genepharma (Shanghai, China). The full length of circPTPN12 with its flanking introns was inserted into a pGCMV/MCS/Neo (pEX-3) vector.

### RNA–RNA pull-down assay

A biotin-labeled circPTPN12 probe and an unlabeled probe (5′-AGGCCATTACAATGATCTGCAA TGAATAC-3′, General Biosystems, Chuzhou, China) were separately incubated with streptavidin-coated magnetic beads (Thermo Scientific), followed by incubation with the lysates of HEK-293T cells transfected with circPTPN12 plasmid. The RNA was extracted from the precipitated magnetic beads and subjected to qRT-PCR, and the resultant qRT-PCR products were further analyzed by agarose gel electrophoresis.

### RNA-binding RIP

The lysates of HEK-293T cells transfected with circPTPN12 plasmid were incubated with Protein-A/G agarose beads (Millipore) and antibody against Ago2 (Abcam, cat# ab186733). RNA was extracted from the precipitates. The abundance of circPTPN12 and miR-21–5 p in the precipitates was detected by qRT-PCR, and the resultant qRT-PCR products were further analyzed by agarose gel electrophoresis.

### RNA FISH

Ishikawa cell sections were hybridized with CY3-labeled circPTPN12 probe (CY3-5′-TTACAATGATC TGCAATGAATA-3′, Geneseed, Guangzhou, China) and FITC-labeled miR-21–5 p probe (FITC-5′-TCAACATCAGTCTGATAAGCTA-3′, Geneseed) at 37˚C for 18 hr. Nuclei were stained using DAPI. The images were acquired with confocal microscopy (TCS SP2 AOBS, Leica).

### Mouse models

All mouse procedures were approved by the Institutional Animal Care and Use Committee at the Nanjing Drum Tower Hospital. BALB/c female mice, at 8–10 weeks of age, 18–20 g, were bred in the Animal Laboratory Center of Nanjing Drum Tower Hospital. Mechanical injury of endometrium was performed as previously reported (*Zhao et al., 2020*). Briefly, mice at estrum defined by vaginal smears were anesthetized with isoflurane. Laparotomy was performed to expose the uterus, followed by insertion of rough surfaced needle to injure uterine endometrium. Mice in sham-operation were just subjected to laparotomy to expose the uterus without injury. AAV-circPTPN12 or AAV-control (10 μl of $1 \times 10^9$ viral genomes/μl) was administered via intrauterine injection 5 days after mechanical injury to study the function of circPTPN12 in endometrium fibrosis. For the therapeutic effect of miR-21–5 p, agomir-21–5 p (Ribobio) or agomir-NC (Ribobio) (5 nmol in 10 μl) was administered every 5 days for three times via intrauterine injection after mechanical injury and AAV-circPTPN12 injection. Mouse model grouping was shown in *Supplementary file 6*. Uterine tissues were collected at estrum, 24–26 days after first surgery. Two side uteri of each mouse were divided into three parts: 1/3 for RNA isolation, 1/3 for immunohistochemistry analysis, and 1/3 for immunofluorescence.

### Statistical analysis

Statistical analyses were carried out by GraphPad Prism software (version 6.0). Student's t-test was used to analyze two experimental groups if data were normally distributed, and one-way ANOVA followed by a Student–Newman–Keuls multiple comparison test was used for comparing three or more groups. Data were presented as mean ± standard deviation (SD) or means ± standard error of mean (SEM) indicated in each figure legend. p<0.05 was considered statistically significant.

## Acknowledgements

We thank Honghong Yao and her team for the technical guidance of circRNA fluorescence in situ hybridization in this study. We thank Yihua Zhou for polishing up the article. This study was supported by the Strategic Priority Research Program of the Chinese Academy of Sciences (XDA16040302), National Key R and D Program of China (2018YFC1004404), National Natural Science Foundation of China (81971336, 81771526, 82071600), Excellent Youth Natural Science Foundation of Jiangsu Province (BK20170051), Jiangsu Province's Key Provincial Talents Program (ZDRCA2016067), and Jiangsu Biobank of Clinical Resources (BM2015004).

## Additional information

### Funding

| Funder | Grant reference number | Author |
|---|---|---|
| The Strategic Priority Research Program of the Chinese Academy of Sciences | XDA16040302 | Yali Hu |
| National Natural Science Foundation of China | 81971336 | Yali Hu |
| National Natural Science Foundation of China | 81771526 | Yali Hu |
| National Natural Science Foundation of China | 82071600 | Guangfeng Zhao |
| Excellent Youth Natural Science Foundation of Jiangsu Province | BK20170051 | Guangfeng Zhao |
| Jiangsu Province's Key Provincial Talents Program | ZDRCA2016067 | Guangfeng Zhao |
| Jiangsu Biobank of Clinical Resources | BM2015004 | Yali Hu |
| National Key R&D Program of China | 2018YFC1004404 | Yali Hu |

The funders had no role in study design, data collection and interpretation, or the decision to submit the work for publication.

### Author contributions

Minmin Song, Data curation, Formal analysis, Writing - original draft; Guangfeng Zhao, Formal analysis, Writing - review and editing; Haixiang Sun, Validation, Investigation; Simin Yao, Zhenhua Zhou, Formal analysis, Investigation; Peipei Jiang, Qianwen Wu, Hui Zhu, Huiyan Wang, Chenyan Dai, Jingmei Wang, Ruotian Li, Yun Cao, Haining Lv, Dan Liu, Investigation; Jianwu Dai, Investigation, Writing - review and editing; Yan Zhou, Yali Hu, Project administration, Writing - review and editing

### Author ORCIDs

Guangfeng Zhao https://orcid.org/0000-0002-9547-0798
Jianwu Dai https://orcid.org/0000-0002-3379-9053
Yali Hu https://orcid.org/0000-0001-5475-7840

### Ethics

Human subjects: This study was approved by the Committee on Human Research of the Nanjing Drum Tower Hospital (No. 2012022), and informed consent was obtained from each participant.
Animal experimentation: All mouse procedures were approved by the Institutional Animal Care and Use Committee at the Nanjing Drum Tower Hospital (No. 2019AE01060). All mouse were bred in the Animal Laboratory Center of Nanjing Drum Tower Hospital. All surgery was performed under sodium pentobarbital anesthesia, and every effort was made to minimize suffering.

Decision letter and Author response
Decision letter https://doi.org/10.7554/eLife.65735.sa1
Author response https://doi.org/10.7554/eLife.65735.sa2

## Additional files

### Supplementary files

- Supplementary file 1. Dysregulated miRNAs in IUA patients.

- Supplementary file 2. Forty upregulated and 40 downregulated miRNAs with higher expression abundance.

- Supplementary file 3. circRNAs with binding sites for miR-21–5 p.

- Supplementary file 4. Clinical information of all patients and controls.

- Supplementary file 5. All primers used in this study.

- Supplementary file 6. Mouse model grouping.

- Transparent reporting form

### Data availability

Sequencing data have been deposited in GEO via the link:https://www.ncbi.nlm.nih.gov/geo/query/acc.cgi?acc=GSE165321. Source data files have been provided for Figures 1, 2, 3 and 4.

The following dataset was generated:

| Author(s) | Year | Dataset title | Dataset URL | Database and Identifier |
| --- | --- | --- | --- | --- |
| Song M, Hu Y | 2021 | Genome-wide analysis of human endometria | https://www.ncbi.nlm.nih.gov/geo/query/acc.cgi?acc=GSE165321 | NCBI Gene Expression Omnibus, GSE165321 |

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
