## [Decision Letter]

**Acceptance summary:**

The authors provide compelling evidence of a circular RNA, circPTPN12, to ultimately induce epithelial mesenchymal transition in endometrium fibrosis, a leading cause of uterine infertility. The authors provide a mechanistic explanation for these findings and elegantly illustrate the physiological relevance thereof in a mouse model of endometrial fibrosis.

**Decision letter after peer review:**

Thank you for submitting your article "Targeting circPTPN12/miR-21-5p/∆Np63α pathway as a therapeutic strategy for human endometrial fibrosis" for consideration by *eLife*. Your article has been reviewed by 2 peer reviewers, and the evaluation has been overseen by a Reviewing Editor and Diane Harper as the Senior Editor. The following individual involved in review of your submission has agreed to reveal their identity: Andrea Romano (Reviewer #1).

The authors performed an impressive amount of work. Experiments and well designed, performed, data are interpreted correctly and overall the quality of the experimental part is excellent, I just have few minor comments (see below). What in my opinion presents important limitations, and should be carefully checked by the authors, is the way data are presented and the discussion, as outlined below.

Essential revisions:

1) The authors cannot justifiably state that this is truly a therapeutic strategy for endometrial fibrosis without formally testing this; the authors should modify the manuscript throughout the text, title and abstract, accordingly. The title implying a therapeutic strategy is in my opinion too assertive based on the data and the Discussion section (see comment 4) of the current manuscript version. Certainly, the term 'targeting' is not appropriate since the experiments were not aimed at 'targeting' this pathway as therapy, but at exploring its involvement.

2) The study design and results are presented as a mix of hypothesis driven and non-hypothesis drive approaches. In the introduction, the aim seems to be hypothesis driven, i.e. the search for post-transcriptional regulators of ∆Np63α. Then in the results, it seems the authors 'interrogate' their transcriptomic data without any supervision or pre-imposed assumption. Was that really the case, or the authors specifically focussed on miRNAS and circRNA with potential interactions with ∆Np63α. The manuscript text should be updated accordingly.

3) In the expression profiling analysis, 40 upregulated and 40 downregulated miRNAs were observed. The authors decided to explore miR-21-5p in particular, because it was expressed and high levels (as relatively high amounts of these molecules are needed to exert their biological functions) and because this miRNA showed strong differential expression between patients and controls (no pre-analyses of potential role of this miRNA with 5p/∆Np63α?). In my opinion, the decision to further explore miR-21-5p is not fully justified: there may other less expressed miRNAs among the 80 differentially expressed that -although less expressed than miR-21-5p, can have a biological effect. In addition, if I look at the heat map in figure 1B, miR-21-5p does not seem to have consistent expression across samples (control2 and IUA1 have basically identical levels of this molecules), and there are other miRNAs with more consistent levels across cases and controls (miR 133, for instance). Similar considerations apply to the selection of the circRNAPTPN12 among those found to be differentially regulated.

4) The discussion does not reach the standards of other parts of the study (especially the results). The discussion is only an extended summary with reference to figures (which in my opinion is not appropriate) and should be extensively re-written. There is too much repetition of Results in the "Discussion" section. It would be more readable if the discussion part would be shortened and reformatted. Since the title suggests the existence of some implications for the treatment of the patients, the discussion should include some possible clinical considerations. What is the relevance of this pathways and p63 in endometrial fibrosis? In other endometrial disorders characterised by fibrosis? It there any data of patient material supporting the implication of this path? What are the clinical and therapeutic scenarios? Reversing fibrosis? Preventing fibrosis?

5) Normally, ΔNp63α is rarely expressed in normal endometrial epithelial cells, the author proposed that the expression of ΔNp63α in IUA is increased due to decreased miR21-5p and increased circPTPN12 as confirmed by WB and luciferase assay, it is very interesting to see whether the expression of ΔNp63α in IUA epithelium is increased by IHC or IF.

6) Whether the localization of ΔNp63α in epithelium would be observed in animal model under different treatment conditions.

---

## [Author Response]

Essential revisions:1) The authors cannot justifiably state that this is truly a therapeutic strategy for endometrial fibrosis without formally testing this; the authors should modify the manuscript throughout the text, title and abstract, accordingly. The title implying a therapeutic strategy is in my opinion too assertive based on the data and the Discussion section (see comment 4) of the current manuscript version. Certainly, the term 'targeting' is not appropriate since the experiments were not aimed at 'targeting' this pathway as therapy, but at exploring its involvement.

We agree to your comments. We are indeed overstatement, so we removed the therapeutic statement from the text, title and abstract, accordingly in this revised version (page 1, Line 1-2; and page 2, Line 37-40; and page 4, Line 74-76; page 13, Line 273).

2) The study design and results are presented as a mix of hypothesis driven and non-hypothesis drive approaches. In the introduction, the aim seems to be hypothesis driven, i.e. the search for post-transcriptional regulators of ∆Np63α. Then in the results, it seems the authors 'interrogate' their transcriptomic data without any supervision or pre-imposed assumption. Was that really the case, or the authors specifically focussed on miRNAS and circRNA with potential interactions with ∆Np63α. The manuscript text should be updated accordingly.

You are right and there were 'interrogate' description in previous manuscript. In this revised version we revised the description accordingly (page 4, Line 80-81; page 5, Line 95-99; page 10, Line 203-204).

3) In the expression profiling analysis, 40 upregulated and 40 downregulated miRNAs were observed. The authors decided to explore miR-21-5p in particular, because it was expressed and high levels (as relatively high amounts of these molecules are needed to exert their biological functions) and because this miRNA showed strong differential expression between patients and controls (no pre-analyses of potential role of this miRNA with 5p/∆Np63α?). In my opinion, the decision to further explore miR-21-5p is not fully justified: there may other less expressed miRNAs among the 80 differentially expressed that -although less expressed than miR-21-5p, can have a biological effect. In addition, if I look at the heat map in figure 1B, miR-21-5p does not seem to have consistent expression across samples (control2 and IUA1 have basically identical levels of this molecules), and there are other miRNAs with more consistent levels across cases and controls (miR 133, for instance). Similar considerations apply to the selection of the circRNAPTPN12 among those found to be differentially regulated.

We focused on the miRNAs targeting ΔNp63α, and miR-21-5p is the highest expression abundance. As the reviewers said, there are other miRNAs and circRNAs that may be involved in endometrial fibrosis but we were not studied. We added this point as a limitation (page 14, Line 302-305).

4) The discussion does not reach the standards of other parts of the study (especially the results). The discussion is only an extended summary with reference to figures (which in my opinion is not appropriate) and should be extensively re-written. There is too much repetition of Results in the "Discussion" section. It would be more readable if the discussion part would be shortened and reformatted. Since the title suggests the existence of some implications for the treatment of the patients, the discussion should include some possible clinical considerations. What is the relevance of this pathways and p63 in endometrial fibrosis? In other endometrial disorders characterised by fibrosis? It there any data of patient material supporting the implication of this path? What are the clinical and therapeutic scenarios? Reversing fibrosis? Preventing fibrosis?

We have deleted the repetition of “results” in the "Discussion" section and have reformatted the discussion part (page 13-14, Line 267-305).

5) Normally, ΔNp63α is rarely expressed in normal endometrial epithelial cells, the author proposed that the expression of ΔNp63α in IUA is increased due to decreased miR21-5p and increased circPTPN12 as confirmed by WB and luciferase assay, it is very interesting to see whether the expression of ΔNp63α in IUA epithelium is increased by IHC or IF.

The expression of ∆Np63α in IUA epithelium is increased by IHC in our published paper and results are quoted in this paper (Zhao et al., 2020).

6) Whether the localization of ΔNp63α in epithelium would be observed in animal model under different treatment conditions.

We did try to detect the protein of ΔNp63α in epithelium with immunochemistry and immunofluorescence, using three antibodies (CST, cat# 67825 and 39692; Abcam, ab124762). Unfortunately, we did not obtain positive results.